# Management and Regulation of Drone Operation in Urban Environment: A Case Study

**Thuy-Hang Tran** [1] **and Dinh-Dung Nguyen** [2,*]

1   Department of Private International Law, Faculty of International Law, Hanoi Unviersity of Law, Hanoi 100000, Vietnam
2   Department of Aircraft System Design, Faculty of Aerospace Engineering, Le Quy Don Technical University, Hanoi 100000, Vietnam
*   Correspondence: dungnd@lqdtu.edu.vn

**Abstract:** With the exponential growth of numerous drone operations ranging from infrastructure monitoring to even package delivery services, the laws and privacy regarding the use of drones in the urban planning context play an essential role in future smart cities. This study provides a comprehensive survey of the regulation of drone application and drone management systems, including a comparison of existing rules, management methods, and guidelines in drone operation to guarantee the safety and security of people, property, and environment. Evaluating existing regulations and laws practiced worldwide will assist in designing drone management and regulation. In Vietnam, the current rules can manage and regulate general guidelines of drone operations based on prohibited, restricted, and controlled areas within the urban context. The legislation, however, is unclear as to how it regulates smaller civilian unmanned aircraft used in the country. In the legal aspect, the potential consequences consist of the inefficiency of compensation responsibility, the violation of drone regulations, and information insecurity.

**Keywords:** UAV/drone management; UAV/drone regulation; UAV/drone application; UAV/drone package delivery; smart cities

## 1. Introduction

Drones, generally unmanned aerial vehicles/systems—UAV/UAS—were initially only utilized for military applications. However, they have recently been widely used in civil applications in many domains such as disaster management (Mohd Daud et al. 2022), the delivery of goods and information (Henderson et al. 2019), search operations (Kyriakakis et al. 2021); (Martinez-Alpiste et al. 2021), surveillance (Dilshad et al. 2020), managing wildfires (Bailon-Ruiz and Lacroix 2020), relay for ad hoc networks (Park et al. 2018); (He et al. 2020), wind estimation (Meier et al. 2022), civil security (Hildmann and Kovacs 2019), agricultural and remote sensing (Zhang et al. 2021), traffic monitoring (Khan et al. 2020), healthcare and health-related services (Kim et al. 2017); (Comtet and Johannessen 2021); (Hiebert et al. 2020), and emergency medical services (Pulsiri and Vatananan-Thesenvitz 2020). Moreover, one of the most evolving areas of drones is their involvement in smart cities, where society and policymakers are continuously working on their developments (Bohloul 2020). The concept and definition of smart cities have gradually evolved since they were first proposed in the 1990s (Albino et al. 2015). The number of publications regarding this topic has considerably increased since 2010, after the appearance of smart city projects and support by the European Union (Jucevičius et al. 2014).

Drones are flexible and fast mobile platforms that can be used for many applications in smart cities, such as traffic and crowd monitoring, environmental monitoring, civil security, and merchandise delivery. They are more flexible and can operate in various locations and situations, including those that are challenging or pose high risks for humans. They can also fly very close to target objects, which allows for higher measurement accuracy and

better-targeted actions. These features provide advantages if drones are used for smart city applications. Due to the demands of residents in smart cities, delivery services must be quicker and more efficient. One of the active solutions to solve this problem is the use of drone applications for delivering packages. With their advantages, drones are already being used in smart cities to improve city life by documenting accident scenes, supporting first responder activities, and monitoring construction sites (Vattapparamban et al. 2016).

Today, there is a large variety of capable drone systems available. However, many systems that meet the payload, range requirements, and environment perspective are light-weight electrical systems which have been improved as a result of battery technology advances. For instance, Amazon and Walmart have been working on a new platform that uses drones to deliver shipments to customers. In 2018, Amazon showed videos of an Octocopter with a flight time of 30 min, delivering packages up to 2.3 kg (five pounds) (Amazon 2018), which is 86% of the deliveries of Amazon. Similarly, DHL, Germany and China's largest mailing company, have started experiments with a fleet of drones that could deliver around 500 parcels per day. The use of drones for daily consumer-oriented services is expanding and becoming a reality (Microdrones 2021).

However, the larger the number of operating drones, the higher the risk of accidents in the sky. This not only endangers civil aviation regarding security and infrastructure but also decreases traffic safety. Recently, some of the most notorious drone-related accidents have been reported (Wilshire Law Firm 2015). Moreover, numerous cases have been reported in which a drone almost collided with an airplane. There was over 700 drone near-miss incidents reported by pilots between January and August 2015, according to data recorded by the FAA (Whitlock 2015). In addition, an investigation of a dataset of drone accidents and incidents in Australia was presented by Milad and Mojtaba (Ghasri and Maghrebi 2021), which showed that two categories of the accidents and incidents are equipment issues and a lack of coordination between aerial activities. Therefore, managing and regulating drones in traffic flows and urban areas is necessary.

The goals of this study are to:

- Review current drone management systems in smart cities.
- Provide a comprehensive investigation of the regulation of drones in some countries.
- Analyze regulations for drone management in Vietnam.

This paper is organized as follows: the authors present a comprehensive review of drone management systems in Section 2. Then, the investigation of drone regulations will be presented in Section 3. Section 4 shows the drone management and law in Vietnam, and the discussion on that issue is given in Section 5. The conclusion is presented in Section 6.

## 2. Drone Management Systems

In this study, a drone is an unmanned aerial vehicle (UAV), which is an aircraft without any human pilot, crew, or passengers onboard. UAVs are a component of an unmanned aircraft system (UAS), which include adding a ground-based controller and a system of communications with the UAV (Hu and Lanzon 2018); (Sharma et al. 2020). The flight of UAVs may operate under remote control by a human operator as remotely-piloted aircraft, or with various degrees of autonomy, such as autopilot assistance, up to fully autonomous aircraft with no human intervention provision (ICAO 2011).

Today, different types of drones have evolved from the advancement in the miniaturization of electronic components, such as sensors, microprocessors, batteries, and navigation systems (Floreano and Wood 2015). A wide variety of drones have been used for military and civilian purposes. Drones range in size from vast fixed-wing UAVs to smart dust, consisting of many tiny micro-electro-mechanical systems, including sensors or robots (Hassanalian and Abdelkefi 2017). A comprehensive list of drones, including airborne and satellite platforms and low- and high-altitude UASs is presented in (Everaerts 2009).

According to the Federal Aviation Administration (FAA), all small drone users are required to register their drones if the weight of the drone is between 0.55 and 55 pounds (FAA 2020). All drones must be managed and controlled as aircraft according to the

FAA regulations (Vattapparamban et al. 2016), which are categorized as the following: (i) model aircraft operators; (ii) those holding 333 exemptions; (iii) public operators; (iv) public operators can choose to operate under Part 107. However, airspace and equipment approval can be achieved depending on who you are and how you want to fly. This procedure is more straightforward and more flexible when operating commercial drones. In addition, a "phased-in approach" has been carried out to integrate drones with a national airspace system, building from rural to urban areas and from low-density airspace to high-density airspace. Moreover, drones can be used for specific applications in restricted areas, for example, atmospheric research.

In European countries, while the Joint Aviation Authorities (JAS) are responsible for operations and licensing, the European Aviation Safety Agency (EASA) is responsible for regulating airworthiness and maintenance issues (ICAO 2017); (EASA 2015a, 2015b, 2016).

In the aviation industry, especially in air traffic control, the increasing number of drones poses new challenges, which endanger the flights operated in the airspaces. Therefore, a unique operating aspect has to be created to protect regular flights regarding safety and fulfilment. Sandor proposed UAS traffic management (UTM) which supports the accomplishment of a flight to manage the total air traffic efficiently (Sándor 2019). Such a system can be used to keep the separation between UAVs and conventional aircraft as well as the order in the traffic flow in the very low-level airspace segments. This system is operated independently of the Air Traffic Management (ATM) due to the data coming to this system.

Based on the scientific results obtained from the creation of an extensive road network model, Péter and Szabó proposed a new air traffic model (Péter and Szabó 2012), which is a powerful new tool in the field of air traffic network modeling and has all the specialties. Such a system is considered an extensive stochastic dynamic system that would be used to describe the processes of land traffic. Because the traffic network is complex and characterized by different rules, geometric data, and seasonality, the authors developed a new model for the air traffic system. The development of a new model became possible, leading to non-linear systems in mathematics. The result from the control point of view was shown by applying the Lyapunov function method, which showed that in a domain bounded by an arbitrary closed curve, the autonomous system is asymptotically stable. The authors used static route parameters to achieve minimized delays in the field of air traffic control. The model used in this research is a macroscopic model based on a linear time-invariant homogeneous differential equation system. The optimization objective of this model is the minimization of total delay; for example, the landing of the aircraft should take place in the shortest time, utilizing the appropriate choice of control parameters. Thus, the solution to significant network problems and the application of new control options are obtained. For instance, it would sometimes be the case that two aircraft are flying close along the same route and planning to travel at the same altitude. In this case, the air traffic controller who has to follow procedures to keep the required separation offers level change primarily. If it is agreed upon, which is usually accurate, there is no need for speed restriction. In contrast, if it is not agreed upon, it must accept the controller's orders.

Another scientific report provided a summary of drones' applications for managing transportation (Barmpounakis et al. 2016). The authors focused on the theory and practice of Unmanned Aircraft Systems (UASs) in transportation and traffic engineering. This survey indicated that the use of drones in transport must guarantee the safety and effectiveness as well as energy-efficiency aspects.

Syd Ali proposed an architectural framework for UTM based on the definition of the UTM system and its six main envisioned functionalities and Communication, Navigation and Surveillance (CNS) technologies supporting the UTM system (Syd Ali 2019). The UTM system is defined as a research software application prototype that aims to safely and efficiently enable UAS operations in low-altitude airspace (Kopardekar et al. 2016). The process of this UTM system is based on: (i) allowing UAS operators to submit flight plans to execute a specific task; (ii) determining how to safely enable single or multiple

UAS operations either within visual line-of-sight (VLOS) or beyond visual line-of-sight (BVLOS); and (iii) coordinating airspace services across many operators. This study also provided six primary functions of the envisioned UTM, which included airspace flight plan processing, operation and management, wind and weather integration, congestion management, separation management, and contingency management.

In recent years, the interest in drones has increased among commercial entities and recreational flyers. Therefore, there is a need to ensure safety for people, properties, and also to other airspace users such as helicopters during drone operations. Pathiyil et al. evaluated the height limit for drone operations in urban airspace with the available technology enablers, categories of drones, and the purpose of the application (Pathiyil et al. 2016). Then, they introduced drone lanes/tunnels or routes concepts, which are enabled for safe drone operation in urban areas. Figure 1 illustrates possible passageways for drone operation, whether it is followed along with water bodies around the area or along the shoreline of the downtown area. These passageways do not affect the existing helicopter operations, disrupt transportation services, nor intervene with industrial and maritime activities.

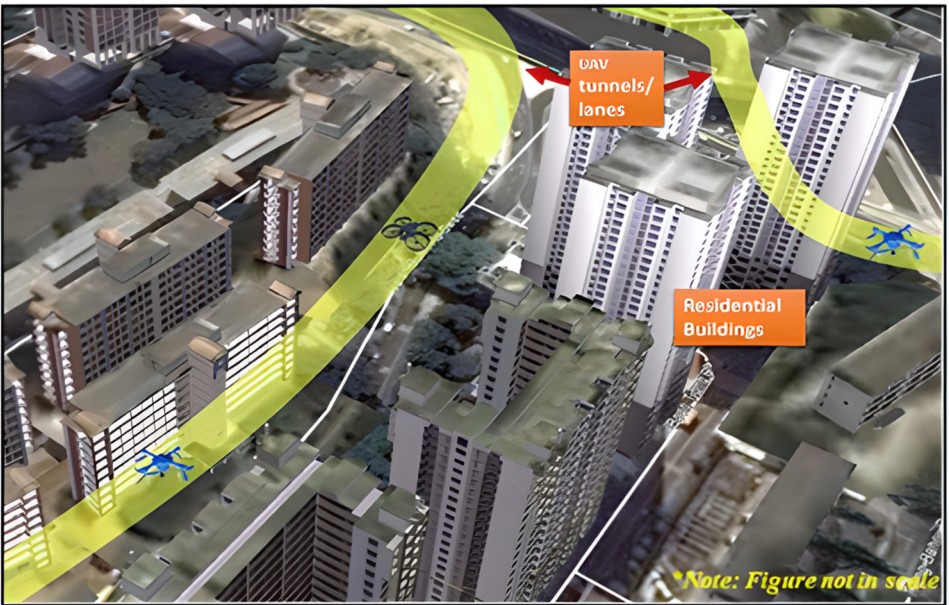

**Figure 1.** Possible drone lanes/tunnels over high raised urban residential areas (Pathiyil et al. 2016).

The routes or lane/tunnels for drone operations over the urban environment are demonstrated in Figure 2. These routes are above or adjacent to land transportation infrastructures in the modern metropolitan cities while keeping a safe distance from moving vehicles and nearby buildings.

To prevent a drone from flying into the safety areas such as airport areas, military installations, or other restricted areas, "Geo-fencing" was introduced and identified. The Geo-fencing methodology provides a virtual boundary for the geographical regions which the drones are not supposed to fly over. This technology remotely oversees geographic areas surrounded by a virtual fence (Geo-fence) and involuntarily detects the mobile objects that enter or exit these areas.

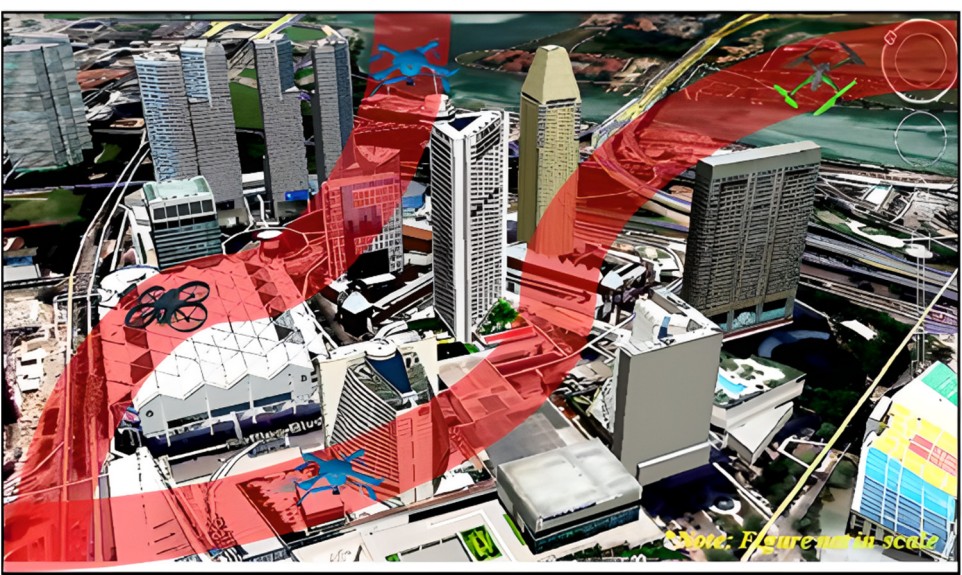

**Figure 2.** A city skyline with designated drone lanes (Pathiyil et al. 2016).

Figure 3 demonstrates the Geo-fencing of some protected areas in a city area such as a government building and an airport area. Drones are not allowed to fly near or enter the airspace of these areas. If they try to get into these areas, the Geo-fencing technology prevents their intrusion by forcing the drones to take a new path or to land outside these areas by a UAV traffic controller, who takes over the control of the aircraft.

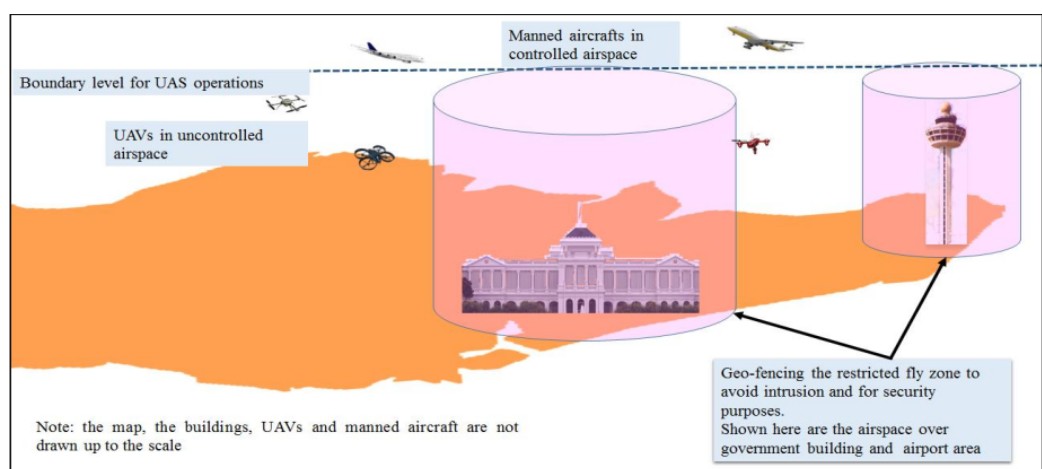

**Figure 3.** Geo-fencing of critical airspace (Pathiyil et al. 2016).

However, this study is considered as an initial step for enabling drone operation in urban airspace, with safety and reliability as the first features.

An approach for managing drones in a smart city is based on cloud devices and services such as computation, storage, and web services (Nguyen 2021). This method is known as an Internet-of-Things-connected UAV. However, operating a drone through the Internet may cause harm to a drone or cause it to crash because of a missing authority or request delay. The best solution to avoid this situation is to use an intelligent onboard device.

Another framework for managing drones in urban airspace called AirMatrix was proposed by Low Kin Huat (Low 2017). This approach is based on a multi-layered altitude and airblocks in latitude and longitude. Figure 4 shows a possible framework for managing drones in a city, including four layers classified by altitude. The first and second layers are above buildings, which are used for high-speed drones operating at heights of 600 ft and 500 ft. The third and fourth layers are used for drone applications in city airspace, which are 300 ft and 200 ft altitudes, respectively. When drones fly at low altitudes, they will

follow selected waypoints, in which drones can change the height to change direction. This framework can allow: (*i*) UTM operators to study and design different airspace structures; (*ii*) the extraction of essential geometric data for analytical studies; the simulation of the traffic of drone operations; and the analysis of airspace performance, such as capacity, efficiency, and safety.

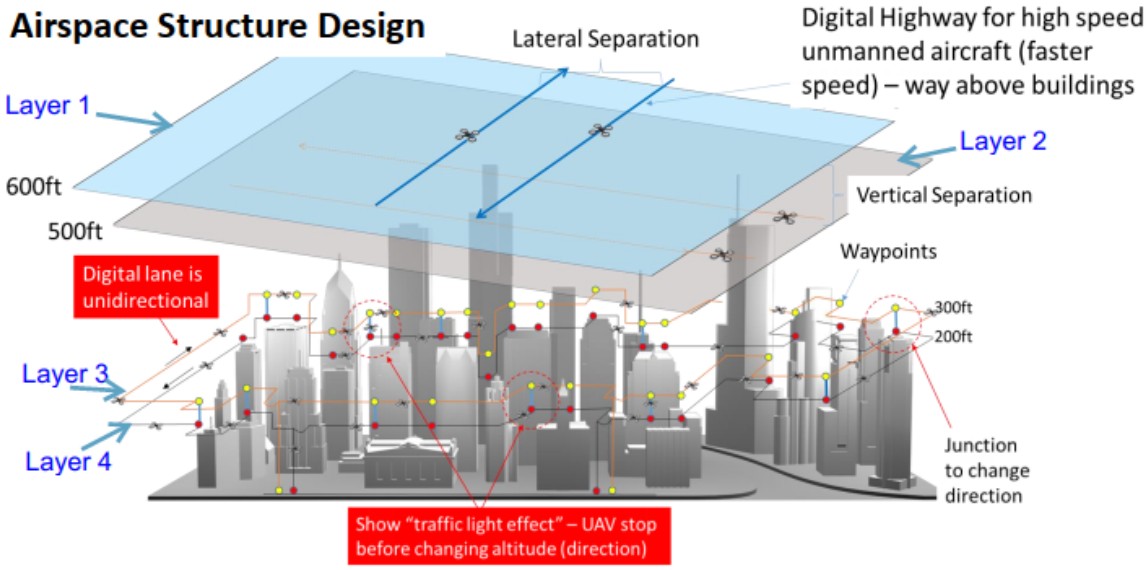

**Figure 4.** A framework structure for urban airspace management (Kin Huat Low 2017).

In the abovementioned literature reviews, it can be noted that the management of drones is a significant issue not only in the transportation system but also in the air traffic system in smart cities. Firstly, the transport system is a whole complex system, the effective operation of which becomes an essential task, which can be widely observed, analyzed, and managed by using an extensive distribution network of sensors and actuators integrated into a system communicating through the internet. Secondly, aerial transportation will continue to increase and face new challenges such as a need for more capacity, more efficiency, and more safety. Thus, the hottest topic of the integration of UTM with a total transport-managing system is the management of drones in urban areas. The primary identified problems are the difficulties of passive surveillance, possible very high traffic intensity, and conflict detection and resolution, including conflicts with built obstacles. The solutions for these problems require the full integration of UTM into the urban transport management systems and the development of unique methods for managing a large number of vehicles in formation flight (Nguyen et al. 2020).

### 3. Drone Regulations

Along with the development of science and technology, drones (UAV/drones) have developed diversely, are widely applied, and are used in many fields, bringing many socioeconomic benefits, such as improving the agriculture sector (Ayamga et al. 2021); (Yawson and Frimpong-Wiafe 2018) and decreasing labor costs (Aurambout et al. 2019). However, there are also potential risks affecting national defense, security, flight safety, and social order and safety, as follows:

(i) *Security and defense.* UAVs/drones can be a means for reactionary forces and opposing forces to take advantage of terrorist activities, surprise attacks on important and sensitive targets, and the unauthorized collection of video recordings. For example, in Vietnam, UAVs have been found encroaching on a military area three times, including the Ministry of Defense (2016); the 3rd Corps, Navy Corps (2017).

(ii) *Aviation safety.* The launch and release of UAVs near airfields where civil and military airplanes fly may seriously threaten flight safety, especially if small UAVs are launched



when aircraft take off or land, which could create an aviation disaster. According to the Report of the Vietnamese National Aviation Security Committee 2019, civil aviation aircraft have collided with an unknown object (most likely a small UAV) which has tipped the nose of the plane, seriously threatening flight safety.

(iii) *Social security, order, and safety.* With a weight from a few hundred grams to several tens of kilograms and with the ability to fly with altitudes from one hundred meters to thousands of meters, in the case of an incident, it could cause danger to people, vehicles, and work. Drones can collect information and images of organizations or individuals illegally.

Some preliminary studies have already focused on drone regulations which impact behavioral privacy (Clarke 2014), regulatory compliance (Huang et al. 2021), perspective from users (Henderson 2022), and public safety (Clarke and Bennett Moses 2014), as well as data protection and ethics (Finn and Wright 2016); (Jansen 2015). In addition, several researchers have focused on the main factors affecting drone policy compliance (Chen and Huang 2021) and safely integrating drones into the national airspace system (Huang et al. 2021). Several analyses and reviews have examined legislations for drone use in specific countries, such as India (Srivastava et al. 2020) and Organization for Economic Co-operation and Development countries (OECD) (Tsiamis et al. 2019).

Henderson evaluated the perspective of New Zealand drone users regarding safety regulations for drones (Henderson 2022). This study found that the current regulatory system is appropriate, with a minority of participants presenting areas for stricter or less strict regulations. New Zealand does not require any pilot qualifications to undertake operations, except for flying within 4 km of an airfield. In addition, a question was raised as to whether the distinction between drones and model aircraft under the rules should be defined by organization membership or aircraft characteristics. However, this study used broad terms due to its largely qualitative approach, and it could have presented issues regarding how participants may interpret some terms, such as regulation or drone.

Dung et al. presented the regulations for drones and proposed drone-following models to manage drones in the urban environment (Dung 2020); (Dung and Rohacs 2018). A higher risk of accidents will occur when more drones are active in the sky, threatening urban air transport such as infrastructure (buildings, public areas) and a safe environment (Wild et al. 2016); (Ren and Cheng 2020). Thus, the need to manage drones in urban areas is necessary. These studies proposed a novel approach to managing drones called the drone-following model, describing the one-by-one following of drone vehicles in urban air transport. This approach was based on defining drone acceleration which depends on the variations in velocities and gaps from the given drone to its front one. Based on the numerical simulation results, the safe distance between drones was kept, which means no traffic flow accidents occurred. Although the numerically simulated results illustrate that proposed models can improve the powerful simulation technologies or a new prototype of controls, drones' equations of motion must be integrated into these models to improve the proposed method.

Fedorko presented a case of studies regarding the legal use of drones (Fedorko et al. 2018), which took place in the Slovak Republic. Generally, the increase in the use of drones will generate a new competitive environment for operating companies as well as cooperative enterprises. However, the recent legislation regarding drones' applications does not satisfy this challenge because of the flexible legal use of drones, such as monitoring employees and delivering documents.

Stöcker et al. presented the highlights of drone regulations (Stöcker et al. 2017), which included a global overview and discussion of primary criteria. Based on the data utilized and analysis techniques applied, the author provided the perspective of past, present, and future trends of the status of drone regulations. In this way, privacy, data protection, and public safety were investigated and discussed in the legal frameworks for operating drones.

Here, we present an evaluation of existing laws and regulations practiced in countries worldwide and designed drone zoning, which can manage and regulate zoning

drones based on banned/prohibited, restricted, and allowed/controlled zones within the urban context.

In Canada, a regulation for operating UAVs set a clear line between commercial use and model aircraft. The weight limit for drones under the category of "model aircraft" is 35 kg; they are individually owned, not under any company for commercial use, and are not used for profit (Transport Canada 2020).

In Singapore, the Civil Aviation Authority of Singapore (CAAS) launched an online portal for drone operators' permit applications and activity permits, required for flying drones weighing more than 7 kg for any purpose, business or recreation (Kok 2015). Recreation or research drones do not require a permit if the aircraft's weight is less than 7 kg. However, those who fly drones for business purposes are required to apply for both licenses regardless of the aircraft's weight (CAAS 2020). In addition, an activity permit is required if the UAV uses restricted or dangerous airspaces or areas within 5 km of a military base regardless of operating height. If drones are flown indoors at a private residence or indoor space, and the flying does not affect the general public, no permits are required.

Thailand regulates the drone application into two categories, including sports and research purposes and personal use, for which users are required to secure prior permission and submit a flight plan. An exception is made for drones used by the film industry, considered to be within the latter category rather than under the former category (Barrow 2015).

Meanwhile, after a series of incidents involving hobby and commercial drones, Cambodia took serious legal action. Phnom Penh's City Hall put an official ban on using drones in Phnom Penh's airspace in April 2015 (Parameswaran 2015). Drones are only allowed to fly only with the permission of City Hall on a pre-arranged flight path. Tourists are still allowed to use UAVs outside the city but are advised to exercise caution when flying drones in heavily populated tourist areas such as Angkor Wat.

In present laws, recreational drone users are not required to pass an aeronautical knowledge test, while commercial drone users are required. In addition, common knowledge of emerging technologies, such as 5G networks, robotics, the Internet of things, and artificial intelligence, in which a drone is an example, is not a regulatory requirement. Therefore, recreational drone users may be incompliant due to their lack of knowledge about regulatory requirements for operating drones. Training and outreach are the most effective methods to bridge this knowledge gap (Chen and Huang 2021). In contrast, commercial drone operators must obtain professional knowledge and practice regarding regulations and codes of conduct. Drone users could acquire this knowledge through outreach and information training by institutions, aviation training agencies, clubs, online resources, or self-study. For example, drone clubs in high schools, colleges, or local communities could serve such a role.

Based on the acknowledgment of the related literature, we can conclude that most countries are aware of the development of UAV/drone usage and future risk to humans, so many efforts have been made, although they are yet to succeed overall. Specific regulations for particular areas and purposes should be developed, especially in urban spaces and those that affect human activities.

## 4. Drone Management and Regulation in Vietnam

In Vietnam, the Ministry of National Defense grants flight permission to Vietnamese and foreign military aircraft operating civil flights in Vietnam and to unmanned aircraft. According to the Civil Aviation Authority of Vietnam (CAAV), flying a drone is legal in Vietnam. Every drone flight conducted in Vietnam requires a special flight license that should be submitted to the Department of Defense at least 14 days before the flight (Bina and Francis 2020). Recently, the Defense Ministry required all drone flyers to have a license to take flight for personal or commercial use. The legislation, however, is unclear as to how it regulates smaller civilian unmanned aircraft used in the country, especially Ho Chi Minh City, which will soon become a smart city.

Since 2019, the drone issue has been included in the announcements and directives of the Vietnamese government related to aviation safety and security to oppose the government. In the Notice dated 22 November 2019, the Government Headquarters concluded as follows:

> "*The Government has paid attention to the issue of threats to security and civil aviation safety of unmanned aerial vehicles (UAV) and ultra-light flying vehicles. With the development of science and technology, UAVs and ultralight flying vehicles are being developed diversely and widely applied in many fields, bringing many benefits to socio-economic life. However, their characteristics are compact, easy to transport, easy to control, easy to manufacture, assemble and use, can operate in environmental conditions that are difficult for humans to access, low cost, and is not strictly managed. Therefore, UAVs and ultralight aircraft are also being abused, becoming a threat to the security and safety of civil aviation activities*".

In 2020, in the directive on enhancing management and supervision measures for driven and super-small flight vehicles, the Vietnamese government again emphasized the dangers of uncontrolled drone use.

> "*In the world, there have been many cases of using UAVs and ultralight aircraft carrying weapons to destroy military targets, assassinate leaders, attack important national targets, hostile territories. In Vietnam, many cases of illegal use of UAVs and ultralight aircraft have been detected and handled in recent years, such as Flying without a flight permit or flying into a forbidden area. Opponents used drones to record protest activities and spread them on social networking sites to propagate and incite anti-sabotage activities. Some military units also detected UAVs, and the ultralight aircraft infringed the military area. Notably, the violations in UAVs and ultralight aircraft increased both in number and dangerous nature. Recently, according to a report of the National Aviation Security Committee 2019, a civil aviation aircraft incident collided with an object in space (likely a small UAV) that dented the nose of the plane, seriously threatening flight safety*".

Stemming from the above issues, the Ministry of Defense deems it very necessary and urgent to request the Prime Minister to issue a decision to establish prohibited and restricted areas for UAVs and other ultralight aerial vehicles.

*4.1. Overview of Vietnamese Legal Document on Dorne Management*

To protect critical national targets from aerial threats, the Vietnamese government issued some legal documents for drone management, as described in Table 1.

Table 1 shows that the essential content of existing legal documents directly regulates drone management. The promulgated documents have low legal value, are sporadic, and lack synchronization among existing legal documents. Only one article in Vietnam's civil aviation law (Article 21 Law no. 66/2006/qh11) defines unmanned aircraft. The remaining regulations related to drone management are by-law documents (decision or ordinance or decree or circular).

**Table 1.** Vietnamese legal on drone management.

| Name of Documents | Year of Issue | Vietnamese Authority | Legal Value | Main Content |
|---|---|---|---|---|
| Decision No. 144/2004/QD-BQP | 30 October 2004 | Department of Defense | Decision | This decision focused on prohibited zones, including 04 areas in the center of cities: Hanoi, Hochiminh City, Haiphong City, and Danang City. |
| Vietnam Civil Aviation Law 2006 (Law No. 66/2006/qh11) | 2006, amended 2014, effected 2015 | The National Assembly | Law | 1. Definition of drones. 2. Category: ultralight aerial vehicles include aerostats and aerial models. |
| Ordinance No. 32/2007/PL-UBTVQH11 | 20 April 2007 | The National Assembly | Ordinance | Protection of essential structures related to national security. |
| Decree No. 36/2008/ND-CP | 28 March 2008 | The Government | Decree | Management of drones and ultralight aerial vehicles. |
| Decree No. 79/2011/ND-CP | 5 September 2011 | The Government | Decree | Amending and supplementing several articles of the government's decree No. 36/2008/nd-cp, of 28 March 2008, on the management of drones and ultralight aircraft. |
| Circular No: 20/2014/TT-BQP | 29 April 2014 | Department of Defense | Circular | Promulgating the regulations on management of Vietnam Air Club. |
| Decree No. 32/2016/ND-CP | 6 May 2016 | The Government | Decree | Management of obstacle height control, airspace control, and air defense systems in Vietnam. |
| Circular No. 35/2017/TT-BQP | 12 February 2017 | Department of Defense | Circular | The eligibility for flight; the eligibility and procedure for licensing of drones and ultralight aircraft. |
| Decision No. 18/2020/QD-TTG | 10 June 2020 | Prime Minister | Decision | 1. Establishing prohibited airspace and flight restriction zones for drones and ultralight aerial vehicles. 2. Determining coordinates and placing names on the field. |
| Instructions No. 347/HD-BQP | 26 February 2021 | Department of Defense | Instructions | 1. Instructing the local authorities to determine the number and boundaries of the prohibited zone and the restricted area by using existing documents and materials or additional measurements 2. Instructing the local authorities to establish a map of prohibited and restricted areas. 3. Establishing a database and map of prohibited and restricted areas nationwide 4. Announcement of prohibited and restricted flying areas on the web portal of the Defense Ministry and each province. |

By 2022, the drone management system was likely to condense into four significant fields as follows:

1. *Definition and Category*

Under Article 21 of Vietnam Civil Aviation Law 2006, amended 2014, effected 2015, (Law No. 66/2006/qh11), an unmanned aircraft is an aerial vehicle of which the flight can be controlled or maintained without direct control by an operator onboard.

✓ Ultralight aerial vehicles include aerostats and aerial models of all kinds.
✓ Aerostat means an aerial craft that gains its lift by using a buoyant gas in its gasbags. Aerostats include powered airships and unpowered balloons.

✓ Aerial models include gliders that simulate shapes and type models of airplanes, have engines, and can be controlled by radio or programmed devices; and paragliders and kites, whether or not man-controlled, except traditional kites.

2. *Regulatory Bodies*

Five agencies are involved in drone management, including the Ministry of National Defense, the Ministry of Public Security, the Ministry of Transport, the Ministry of Science and Technology, and the Ministry of Finance. Directive No. 02 requires these agencies to coordinate with each other, but there are no guidelines on coordination mechanisms or the power to propose coordination. The Ministry of National Defense will be the lead agency responsible for prescribing airworthiness standards, criteria, and procedures for licensing establishments to design, manufacture, repair, maintain, or test unmanned and ultralight aerial vehicles and their engines, propellers, and equipment.

3. *Prohibited areas and Flight restricted areas*

Before 2008, drones had not been developed for civil purposes. So, Decision No. 144/2004/QD-BQP, 2004 was mainly aimed at human-crewed aircraft to prohibit aircraft from flying into densely populated areas and important political, administrative, and cultural centers to ensure the safety of people and workers in the area. Decision 144 does not prohibit drones from flying in airport areas, which means that drones can fly in these areas, which is inconsistent with ICAO's recommendations on flight safety and is unsuitable for current law provisions. In the face of recent complicated developments in the use of drones, the decision regarding the prohibited and restricted areas for drones has promptly adjusted the operation of drones to ensure national defense, security, flight safety, and social order and safety. Therefore, in 2020, Decision No. 18/2020/QD-TTg was issued to establish prohibited zones and restricted flying areas for drones and ultralight aircraft.

Decision No. 18/2020/QD-TTg aims to implement Decree No. 32/2016/ND-CP and the Ordinance on Protection of Important Works related to National Security 2007. These two documents define the boundaries on the ground and prohibited acts to protect national works, such as defense and military zones and essential works related to national security; however, the boundaries in the airspace have not been determined and specified for the drone's flight in these areas.

4. *Flight permits*

Based on Circular No. 35/2017/TT-BQP established airworthiness standards and licensing procedures, including design, manufacture, repair, maintenance, and testing of equipment and devices, the standards were defined in Article 3, which consists of parameters and requirements regarding the technical standards of aircraft, avionics, and equipment of drones and ultralight aircraft for maintaining safe and secure operations. However, there is an absence of qualitative standards decided by the government agency. Instead, the eligibility of drones for flights is upon its operator's provision of complete, specific, and accurate information on (i) characteristics for the identification of the drone and (ii) fundamental specifications.

When wishing to organize flight activities, organizations and individuals shall submit dossiers of application for flight licenses (Vietnamese Government 2008). A flight license contains the following required information:

➢ The name, address, and telephone number of the licensed organization or individual.
➢ Identification features of the aircraft type (including an annex containing photos and an explanation of the technical properties of the aircraft).
➢ The zone in which flight activities are permitted, flight direction, and trail.
➢ The permitted purposes, duration, and time of organization of flight activities.
➢ Regulations on the notification of flight coordination; a designated agency to administer, supervise, or manage flights.
➢ Limitations and other security or defense regulations.

In 2004, UAVs had not been developed for civil purposes. So, this decision was mainly aimed at human-crewed aircraft to prohibit aircraft from flying into densely populated areas and important political, administrative, and cultural centers to ensure the safety of people and workers in the area. In the face of complicated developments in the use of UAVs recently, the decision regarding prohibited and restricted areas for UAVs has promptly adjusted the operation of the UAVs to ensure national defense, security, flight safety, and social order and safety. Therefore, in 2020, Decision No. 18/2020/QD-TTg was issued to establish prohibited zones and restricted flying areas for UAVs and ultralight aircraft.

Generally, legal documents are scattered in many laws and guiding decrees. The most important content of existing legal documents is directly associated with regulated drone management.

### 4.2. Prohibitted Areas

According to the International Civil Aviation Organization (ICAO), a prohibited area is restricted airspace defined in the territory of a country in that prohibits flight operations. Vietnamese law already has regulations on no-fly zones in Clause 1, Article 85 of the Vietnam Aviation Law: "Prohibited zone means an aerial area with a defined size where an aircraft is not allowed to enter unless Vietnamese public service aircraft on duty".

Under Decision No. 18/2020/QD-TTg of the Prime Minister, the prohibited zones are determined as follows:

(a) Areas of defense works and particularly important military zones: The distance of the operating UAV to the boundary of the restricted area horizontally must not be less than 500 m at all altitudes.

(b) Head office areas, including Party, State, National Assembly, Government, department, ministry, central branch, central city areas. The distance of the operating UAV to the boundary of the restricted areas is not less than 200 m horizontally at all altitudes.

(c) Areas for national defense and security, including military stationed areas and areas on the list for important works related to national security. The distance of the operating UAV to the boundary of the restricted areas is not less than 500 m horizontally at all altitudes.

(d) Airports and airfields where civil and military aircraft are operated: For an airport, the restricted range is limited to a rectangular area. The rectangular length is 15,000 m extending along the two ends of the runway, and its width is 5000 m extending to both sides of the runway. For airport areas with operations of civil aviation aircraft with a frequency of fewer than six flights per day, flexible prohibited zones are allowed; however, organizations and individuals operating UAVs may fly only after having a close agreement with air traffic control agencies at airports. UAVs may not fly within the boundaries of an airport within one hour before and after the operation time of the aircraft.

Comparatively, recognizing the complexity of UAVs, many countries have issued prohibitions and restrictions on UAV operations, specifically:

(i) In the US, UAVs can only be operated at altitudes less than 120 m above the terrain, cannot operate near airports within about 8 km, and cannot operate near defense and security areas. They are restricted in crowded areas.

(ii) In the UK, UAVs are banned from operating near airports within about 01 km; UAV flight near nuclear power plants, military zones, telecommunications centers, prisons, and pilot training areas is prohibited; UAVs must fly more than 150 m away from a crowd.

(iii) In China, it is forbidden to fly UAVs over 120 m. UAVs cannot fly in densely populated areas, around airports, military zones, other sensitive areas such as police checkpoints, or a controlled area, unless approved by the Aviation Administration.

(iv) It is forbidden to fly UAVs more than 90 m above the terrain in Thailand. UAVs cannot operate near airports within 9 km unless otherwise permitted by the airport operator; they cannot fly over cities, villages, or areas where people gather or near

human-crewed aircraft; they must fly more than 50 m away from people, vehicles, or constructions.

*4.3. Flight Restricted Areas*

As defined by the International Civil Aviation Organization (ICAO) in Annex 11 on Air Traffic Services, a restricted area is restricted airspace defined in the territory of a country, which limits flights under certain conditions. The Vietnam law has specific regulations on restricted zones in Clause 1, Article 85 of the Law on Aviation of Vietnam: "The restricted flight area is an aerial area with a defined size in which an aircraft is only allowed to operate in that area when specific conditions are met.

A restricted flight area is determined under Decision No. 18/2020/QD-TTg of the Prime Minister, as follows:

(a)  The airspace is over 120 m high above the terrain. This area does not include the airspace of prohibited zones.

(b)  Border areas, including: (i) the land border between Vietnam and China is 25,000 m from the border to the interior of Vietnam at all altitudes. (ii) The land border between Vietnam and Laos and Cambodia is 10,000 m from the borderline into inland Vietnam at all altitudes.

(c)  A crowded area: A crowded concentration area is defined as a restricted area, whereby when UAVs operate in a crowded area, they must meet the conditions of the licensing authority, such as: they cannot fly over people; there must be a warning guide to ensure safety in case the UAV has technical problems that can cause accidents; they must use the image information collected from the UAV following regulations to protect the privacy of each person. In addition, the determination of a crowded area as a restricted flight area also aims to ensure security, social order, and safety.

(d)  Areas adjacent to prohibited zones at airports where civil and military aircraft are operated. The defining restricted flight range is extended to 3000 m wide and 5000 m long from the boundary of the prohibited zone at the airport; the altitude is less than 120 m above the terrain.

## 5. Discussion

Smart cities often refer to data-driven urban science and management, IoT devices, smart homes, and smart infrastructure. With advances and innovations in emerging technologies, such as IoT, robotics, artificial intelligence, and 5G networks, smart city scenarios and processes were proposed as various intelligent resolutions to handle city resources and assist residents (Chen and Huang 2021). A drone is an example of emerging technology being integrated into smart cities for varied benefits (Khan et al. 2018), (Mohamed et al. 2020). Moreover, the safe and effective use of these emerging technologies will extend the capacity of intelligent devices in a smart city environment and supply creative resolutions to overcoming limits charged by existing city infrastructure. In addition, drones could be employed in many areas of smart city operations, such as parcel delivery, traffic management, public safety, and infrastructure inspection. However, drone operations raise public policy concerns, such as safety, security, and privacy, which are primary concerns for smart cities. While privacy concerns are considered non-technical issues and are addressed from regulatory standpoints, safety and security are discussed as technical issues for drone operations. Therefore, drone management and regulation development and implementation benefit smart cities by addressing safety and privacy concerns.

In Vietnam, drone management and regulation have fundamental characteristics compared to selected countries' aviation rules. Table 2 compares the most basic aviation rules applied to drones in Vietnam and selected countries.

**Table 2.** The comparison of most basic aviation rules applied to drones in Vietnam and selected countries.

| Basic Aviation Rules | Vietnam | United States | United Kingdom | China | Singapore | Thailand | Canada | Cambodia |
|---|---|---|---|---|---|---|---|---|
| Airfield | Permission required to fly at or near an airport | Permission required at controlled airfield | Permission required to fly at or near an airport | Permission required to fly at or near an airport | Permission required to fly at or near an airport | Permission required to fly at or near an airport | Permission required to fly at or near an airport | Permission required to fly at or near an airport |
| Requirement of pilot qualification | No | Required | Required | Required for some | Required | Required for some | Required for some | Required for some |
| Flight permits | Required | Required | Required | Required | Required | Required | Required | Required |
| Operations near private property | Allowed | Allowed | 150 m horizontal distance from residential, recreational, commercial, and industrial areas | Required for some | Required for some | Required | Required for some | Required |
| Visual line-of-sight | Not required | Required | Required | Required | Required | Not required | Required | Not required |
| Altitude limits | 120 m | 120 m | 120 m | 120 m | 60 m | 120 m | 120 m | 120 m |
| Night flying | Allowed | Anti-collision lighting and test required | Special permission required | Allowed | Allowed | Allowed | Special permission required | Allowed |
| Model flying clubs | Allowed | Special permission under rules | Special permission under rules | Allowed | Allowed | Allowed | Special permission under rules | Allowed |
| Distinction between commercial and recreational | No | Yes | No | Yes | No | No | Yes | No |

Vietnam is a case study because its regulations are analogous to the model regulations from the International Civil Aviation Organization (CAO) and other major jurisdictions (ICAO 2021). From Table 2, the author found that Vietnam's management regulations are less strict than other legal systems. Flight operators are allowed to operate in some ways that may be prohibited in other countries, such as night flying and operation near private properties. Overall, the only management method in those documents is to set up prohibited and restricted zones.

Vietnam does not currently require any pilot license to undertake operations, making Vietnam at odds with significant jurisdictions such as the United States, the United Kingdom, Australia, and Singapore (FAA 2022); (CAAUK 2021); (CASA 2021); (CAAS 2022). However, the Ministry of Transportation in Vietnam has been proposing an essential pilot qualification for drone users who want to have scope for licensing or similar systems to those in other jurisdictions. This could be an interesting area for future research to examine Vietnam's aviation safety regulatory framework for drones from the perspective of drone users.

In these legal corridors, content supports and assists drones' flight operations. Drones are allowed to operate in many fields. According to their registered specifications, operators have the right to decide on the drone management model in the permitted area. There has not been any classification or limitation on the technical characteristics of drones from the government. These conditions make drone applications in Vietnam an impressive development. Drones help Vietnam's agricultural industry digitize effectively, solving the labor shortage in rural areas. Scientists have joined hands to participate in the production and application of drones. Typical in the startup community is Mismart's drone solution, which won the top prize in the "Ho Chi Minh City artificial intelligence application innovation project" contest in 2020. Mismart, a Vietnamese Company, mastered the production technology with a localization rate of 70%; the selling price is about VND 150–250 million/piece—half of that of imports from the US, Europe, and Japan. Simple regulations may have influenced the early stages of promoting and encouraging drone development. However, an effective management model needs to be built in the future to limit possible damages and risks.

It is not clear which management model should be used in permitted areas. Under Circular No. 35/2017/TT-BQP on establishments' airworthiness standards and licensing procedures, including the design, manufacture, repair, maintenance, and testing of equipment and devices, the authorities controlled the operation of UAVs and ultralight aircraft through legal papers carried during operation, not by any of the drone-following models to manage drones in the urban environment.

This remaining problem can lead to possible consequences as follows:

(i) The Vietnamese government has issued management regulations. However, it has not given specific sanctions, leading to the fact that competent authorities often avoid or are confused when dealing with violations of drone owners. If individuals violate the prohibited and restricted fly zones, they can be administratively sanctioned or penalized. Nevertheless, there is no specific regulation on the level of violation they will be given. Competent state agencies of Vietnam shall apply Decree No. 167/2013/ND-CP sanctioning administrative violations in social security, order, safety, prevention of social evils, fire prevention, fighting, and domestic violence prevention and control. The point g, Article 5 stipulates, "*Flying kites, balloons, airplanes, remote-controlled saucer or other flying objects in the airport area, prohibited area; burn and release "sky lanterns""*; with fines from VND 50,000 to 1,000,000 (equal to a maximum of USD 17). The fines are too mild and are not deterrents compared to the consequences drones can cause.

(ii) In the areas permitted, the absence of management guiding rules will lead to the inability to determine the damage or the responsibility to compensate. A drone is not an asset that requires ownership registration like a car or motorbike in Vietnam. When damage occurs, who will pay for the damage? The owners, operators, or jointly? For

example, suppose an accident occurs under Article 601 of the Vietnam Civil Code 2015. In this case, the user and the vehicle owner must jointly compensate, so will such a case apply to drones?

(iii)  There is no agency responsible for the quality control and inspection of drones. Most drones in Vietnam are imported, resulting in the risk of information insecurity. Therefore, UAV operation may send the collected data to a server in the country of manufacture. There are no specific regulations to prevent imported drones from sending data to servers abroad or controlling this activity.

## 6. Conclusions

In this paper, we presented a comprehensive survey on drone management systems and regulations. Some countries control drone operations, and others prohibit drone operations in urban areas. Several methods for managing drone applications were presented, such as drone lanes/tunnels over urban residential areas, drone-following models, AirMatrix, and Geo-fencing, which guarantee safety for people, properties, and other airspace users. As a case study, Vietnam does not require any license operations for drone users and does not give any classification or limitation on the technical characteristics of drones. Therefore, it could be an interesting area for future research to develop Vietnam's aviation safety regulatory framework using advanced IoT, robotics, the 5G network, and artificial intelligence. Based on the investigation and evaluation of the existing laws and management methods, we present drone regulation and management in Vietnam based on prohibited and restricted areas. However, this legislation is still unclear as to how it regulates smaller civilian unmanned aircraft, which raises potential consequences regarding legal aspects, such as the inefficiency of compensation responsibility, the violation of drone regulations, and information insecurity. In legal corridors, content supports and assists drones' flight operations. However, reality proves that these conditions have meant that drone application in Vietnam has developed to an impressive extent.

**Author Contributions:** Conceptualization, D.-D.N.; methodology, D.-D.N. and T.-H.T.; validation, D.-D.N. and T.-H.T.; formal analysis, D.-D.N. and T.-H.T.; investigation, D.-D.N. and T.-H.T.; resources, D.-D.N.; data curation, D.-D.N. and T.-H.T.; writing—original draft preparation, D.-D.N. and T.-H.T.; writing—review and editing, D.-D.N.; supervision, D.-D.N. All authors have read and agreed to the published version of the manuscript.

**Funding:** This research received no external funding.

**Institutional Review Board Statement:** Not applicable.

**Informed Consent Statement:** Not applicable.

**Data Availability Statement:** Not applicable.

**Conflicts of Interest:** The authors declare no conflict of interest.

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
