# Peer review of "Management and Regulation of Drone Operation in Urban Environment: A Case Study"

_socsci, doi:10.3390/socsci11100474_

Round 1
Reviewer 1 Report
Review of the manuscript ‘Management and regulation of drone operation in urban environment: a case study’.
The manuscript aims to contribute to the knowledge of drone regulation and flight controls. The interest for this topic is surging and there is an extensive scientific literature that has emerged the last years.
I I'm not sure what kind of article this is intended to be. The headline states “a case study”. However, the substance of the manuscript is based on references and citations from harvested literature. It has no obvious character of a case report, it has not conducted any “particular study” although there seems to be certain focus on Vietnam using citations from some material from regulatory documents. But what should be “the case study”? It is stated that the paper shall present a “comprehensive survey on drone management systems and regulations”. However, it does not satisfy any of the established criteria for any classes of a review. It is not stated any research strategy or criteria for how the referenced literature is harvested. Is most of the literature harvested arbitrarily, by a snowball strategy or what?
The presented literature is not representing up to date literature reflecting the global extension of drone research and development. As an “overview”, it has an unsupported skewness for Asia. Furthermore, it does not contain much of the most recent literature of the topics related to drone management system, airspace regulation and technology development. A large part of the paper is related to the management of regulation in Vietnam. The character of that specific content is rather dubious. Some of the text related to Vietnam is referring to/citing regulatory documents, some text is more like a characterization of the regulations of Vietnam using tendentious and questionable arguments that there is no supporting substance for in the literature used.
The manuscript mentions some of the purposes that drones are used for, i.e., traffic and crowd monitoring, environmental monitoring, civil security and merchandise delivery. It is amazing that one of the use-cases that has gained the most interest, drone services in healthcare, is not mentioned at all.
Discussion
The “Discussion” is mostly focused on some regulatory issues related to Vietnam, is rather tedious and with details that add no useful information to the drone literature. It is actually remarkable that the discussion exclusively is dealing with Vietnam without relating this to other literature from other parts of the world.
Quality of text.
There is to much content that is unsatisfactory, but as som simple illustrations I quote and comment on some.
Line 53: However, the larger the number of operating drones, the more accidents in the sky. It not only endangers civil aviation regarding security and infrastructure but also decreases traffic safety.
Line 208: More accidents will occur when more drones are active in the sky, threatening urban air transport such as infrastructure (buildings, public areas) and a safe environment.
That more accidents will occur is a speculation in this document. There is no substantial evidence for the statement. It may be true of course, but we do not know. And there are multiple other factors that may influence the number of accidents. This text is not appropriate in a scientific paper.
Line 108: For instance, it would sometimes occur that two aircrafts flying close to each other along the same route want to travel at the same altitude. In this case, the air traffic controller who wants to keep the required separation offers level change primarily. If it is agreed, which is usually accurate, there is no need for speed restriction. If the other flight levels or altitudes are occupied, then the crew may be asked to maintain a certain speed.
“want to travel at the same altitude” and “controller who wants to keep…”. This is not a scientific statement but rather an unformal way of writing, more appropriate for unformal chronicles. And I do not believe controllers may “want” anything, they will have to follow procedures with extremely strict accuracy.
Line 199: The solutions for these problems require the full integration of UTM into the urban transport-management systems and the development of unique methods for managing a large number of vehicles in formation flight (D. D. Nguyen et al. 2020).
This is a well known topic, and it is more interesting to discuss how such solutions should be planned instead of only stating the challenge.
Line 268: Based on the overview, we can conclude most countries are aware of the development of UAV/drone usage and future risk to humans, so that many efforts have been made, although it is yet to succeed overall. The specific regulations for the particular areas and purposes should be developed, especially in urban spaces, and affect human activities.
As I have stated earlier, this is not qualifying for an “overview”. It is based on some 44 “papers” that do not seem to be systematically selected, and it is a mixture of differing sources, many not scientifically conducted.
Line 425: There are too many agencies involved in drone management….
“Too many” is a normative statement. On what background is this stated? There is not presented any data in the paper.
Language and clarity.
The document would need extensive English editing as multiple phrases and statements are not fully understandable with respect to what the authors intend to state.
Conclusion.
This paper does not fulfill crucial criteria for scientific articles, and I cannot see that it adds information of interest. I therefore cannot advise that this paper is accepted for publication.
Author Response
The authors would like to thank the reviewer for the comments, which significantly helped to improve the manuscript. We have addressed all the issues raised in the review. For each comment, we provided a response and also described how the manuscript was revised. Those changes are highlighted within the manuscript. Please see the attached file.

Reviewer 2 Report
The article addresses an issue of great relevance and complexity in the interaction between specific air transport technology and urban development. This concerns the management of drones in the context of the evolution of smart cities.
Being already used in air transport, drones have determined specific regulations (ATU). In a rapid expansion, drones began to be used in commercial, social or, in general, economic activities that interfere with the evolution of smart cities.
New transport networks (lanes, tunnels, layers) have been designed to ensure social security, protection of the environment and the citizen, ethics of personal data, etc ..
The article highlights the framework models of urban airspace management such as AirMatrix, Geo-fencing, etc ..
"The solutions for these problems require the full integration of UTM into the urban transport-management systems and the development of unique methods for managing a large number of vehicles in training flight" states the author.
At the same time, they present preliminary studies on the evolution regarding the complex regulation of the use of drones in urban spaces, in particular. The fundamental objective of these regulations is to assess the impact on public behavior and safety.
The examples used by the authors regarding urban areas in Vietnam and the Asian area come to substantiate the growing interest in the issues discussed.
I propose to accept the article for publication in this form.
Author Response
We thank this reviewer for his/her honest feedback. We also appreciate that the reviewer acknowledges the importance and timeliness of our study. Based on the comments made, we have made an improvement in the revised manuscript.
Reviewer 3 Report
Overview:
The manuscript presented a comprehensive review of UAV policy and regulation for operations in urban environments and closed with a case study in Vietnam. The manuscript is well-written and logically organized, providing practical guidelines, rules, and insights for the management of drone operations in the context of developing smart cities. Overall, the review of the UAV regulation seems to be complete, and I think there are a few aspects that the authors should talk about to make the discussion more holistic, such as the classification and definition of a UAV and the IoT aspect of a drone towards its usage for benefiting a smart city. I only have a few comments.
Comment 1: Line 67-68 review different regulations (e.g., FAA, JAS) for UAV operations and flying areas. I would suggest the authors provide clear definitions of UAVs and different types of UAVs in the context of these regulations. The UAV regulations, such as the one from FAA, are modified and evolving throughout time. The definition of UAV also varies, which could significantly affect the accuracy of this manuscript. As an example, in the past few years, unmanned aircraft that are lighter than 250 grams (such as the DJI mini-series) or have other physical attributes are not considered a UAV by the FAA regulation. Therefore these small planes are not subjected to UAV regulations and rules. However, they are still drones and can be used for research and urban management operations. A clear definition of the UAVs, their classification, and types (e.g., battery-based vs. fuel-based, commercial vs. recreational, and different flying altitudes) is needed, as the regulations these drones are subjected to are probably different.
As a controversial example, many research institutes use unmanned airships and dirigible balloons to conduct topographic surveys or take air pollution samples, are these unmanned aircraft considered UAVs or UASs? They probably are subjected to a completely different set of rules that regulates quadcopter drones.
Comment 2:
Line 168-171 ~ “An approach for managing drones in a smart city is based on cloud devices, and ser-168 vices such as computation, storage, and web services were presented by (D.-D. Nguyen 169 2021)”, I think the formal name for this is “Internet of Things (IoT)-connected UAV”.
Comment 3:
An important aspect of “UAV courses, pilot licenses, and mandatory training” is missing from the current manuscript. Based on my best knowledge, in many countries, UAV pilots are required to take classes/training/exams and get their certificate for flying a drone for both commercial and recreational purposes. This aspect should be a critical part in reviewing UAV regulations.
Comment 4:
I would also encourage the authors to discuss how UAV operations can benefit smart cities (as the term is mentioned in the manuscript). UAV for smart city operations is probably a novel topic, which differs from past UAV applications that require manual control of drones from a pilot. Smart cities often refer to data-driven urban science and management, IoT devices, smart homes, and smart infrastructure. How UAVs can play important roles in these “Smart” contexts can add value to the current manuscript.
Comment 5:
The conclusion is too short and should have presented unique insights. A good review paper should offer some useful insights and opinions of the authors to shed light on the future research and development of the domain/subject area. That would also be counted as the unique contribution of the manuscript. I think the author could present many insights, such as the regulation and knowledge gaps in the existing UAV management system in Vietnam, and propose potential solutions (using IoT, smart city technology, the body of knowledge, 5G network, or UAV-based edge computing) to address these gaps.
Author Response

(The authors gave the same response as above.)

Reviewer 4 Report
This paper as the authors write (line 166) "is considered as an initial step for enabling drone operation in urban airspace with safety and reliability as the first feature".
In my opinion, this is a review and not a research paper "article".
The paper does not contain original research but only a review of existing regulations and their commentary.
This work should change type to "review" and and then go through the review process.
Comments:
1. Line 187: Figure 4 is authors' picture or from paper (Kin Huat Low 2017)? This is not explicitly clear from the text of the work. If it is an image from a cited work then refer to it behind the figure caption, such as in Fig. 3 caption.
2. Chapter "Conclusion" is very general and obvious. Due to the lack of scientific research, there are no contributions influencing the development of this field.
Author Response

(The authors gave the same response as above.)

Reviewer 5 Report
Overall, the manuscript is easy to read, and structure is clear, however, the following issues were observed, and the author(s) are encouraged to improve quality of article.
1. Language errors, typos, and confusing sentences are observed across the manuscript,
For example:
The KeyWords, “smart citie”.
The second sentence on Page 2, what is the 3x30 minutes for the flight time? Explanation is needed.
The first sentence in Section 2, page 2, “According to the Federal Aviation Administration (FAA), the area for operating drones is classes of activities or specific operations that are usually prohibited for commercial drone applications.” This sentence needs rewording, I can hardly understand. In addition, the in-text cite should be the original source from the FAA as the author(s) state “according to the FAA”.
What is the “phase-in approach”? the author(s) brought in the concept but no explanation for readers.
2. This manuscript does appear to be a typical case study, but it more looks like a piece of study review. Any explanation that authors could offer?
3. In the Section 3 “Drone Regulations”, many more recent studies have been published related to drone regulation compliance, policy and privacy, which should be considered. For example, Huang, C., Chen, Y-C, & Harris, J. (2021). Regulatory compliance and socio-demographic analyses of civil unmanned aircraft system users. Technology in Society, 65., Chen, Y-C, & Huang, C. (2021). Smart Data-Driven Policy on Unmanned Aircraft Systems (UAS): Analysis of Drone Users in U.S. Cities. Smart Cities 2021, 4(1). Henderson, I. L. (2022). Aviation safety regulations for unmanned aircraft operations: Perspectives from users. Transport Policy 125(2022).
4. The most important observation and suggestion for improvement: How do the three goals of study collectively contribute to the body of knowledge or future studies? Does this manuscript specifically target the case in Vietnam? If so, what knowledge, information, or suggestions could readers from other countries can take away? The author(s) are encouraged to think more and add additional discussion from that perspective.
Author Response

(The authors gave the same response as above.)

Round 2
Reviewer 1 Report
Second Review of the manuscript ‘Management and regulation of drone operation in urban environment: a case study’.
I observe that the authors have responded to some of the previous topics. Unfortunately, I still find several challenging elements in this draft.
The “comprehensive survey” has been modified and extended to some degree. However, I still find it in lack of relevant literature, and some of the referred papers are not sufficient as they are dated some time back.
As an example; in lines 55-62, the referred reports of danger and accidents are from 2015. This was before most of the current regulations that are implemented in many systems today (i.e., US/EU).
Lines 242-244: More accidents will occur when more drones are active in the sky, threatening urban air transport such as infrastructure (buildings, public areas) and a safe environment (Wild, Murray, and Baxter 2016), (Ren and Cheng 2020).
I do not accept the statement “the larger the number of operating drones, the more accidents in the sky”. This has been disproved by civil air traffic. The number of airplanes and flights have surged the last decades, but the number of accidents has not increased thanks to the continuous learnings and improvements from accidents that have taken place. It is very well accepted that the RISK of accidents will obviously increase with increasing number of UAVs in the airspace. However, it is the goal for developing regulations and drone steering systems to prevent such problems related to drone transports.
There are still too many linguistic flaws in the text. Examples of some sloppy text:
Lines 129-134: For instance, it would sometimes occur that two aircraft (should be aircrafts) are flying close along the same route and desire (Who may desire? Pilot? Aircraft? They may not desire, they have to obey instructions) to travel at the same altitude. In this case, the air traffic controller who has to follow procedures to keep the required separation offers level change primarily. If it is agreed (they may not “agree”, they must accept the controllers orders), which is usually accurate, there is no need for speed restriction. In contrast, the crew (in the future, it will probably fly autonomously) may be asked (they will be told to = ordered ) to maintain a certain speed if the other flight levels or altitudes are occupied.
Other examples of unprecise formulations:
Line 184: “This study”…”. Is it referring to the current submitted study, or the study of Pathiyil, or a report of Geo-fencing?
Line 30: A smart city is an urban region that is highly advanced in terms of overall infrastructure, sustainable real estate, communications, and market viability.
I think this first introduction of smart city is rather narrow. It is partly improved later, but should be stated more clearly when smart city is used for the first time in the document.
Line 35: Unlike manned planes, drones can be more cost-effective.
This statement needs documentation. I am not aware of scientific reports proving that drones in general are more cost-effective. It is argued to be so, but most proposed economic models have not included all costs that will develop with more regulations, i.e., air space costs related to surveillance, communication costs and more.
The authors still focus heavily on package delivery. They have included a citation for health care relevance (Comtet et al). That study had a very narrow perspective and is not covering the huge potential in health services, but there is a rich literature that may be used.
Line 295-313: This is related to Vietnam at the end of a chapter from other systems. It may possibly be better fitted into the next chapter; 4. Drone management and regulation in Vietnam.
In contrast, lines 321-339 is mostly general topics of drones as I see it. It is not specifically described as issues related to Vietnam. At least, it is applicable for most systems. Part of it should be integrated with the previous sections.
Line 302-303: In present laws, recreational drone users may not be required to pass an aeronautical knowledge test, while commercial drone users are required.
How to interpret? I understand this as it is given by law that it is not allowed to demand that recreational drone users shall pass a test. Or is the intention to tell us that it is a lack in the system that they are not obligated to do so? Unclear to me.
4. Drone management and regulation in Vietnam.
Some examples of unclear statements. There are many.
Lines 322-323: “…bringing many benefits to socioeconomic”. Socioeconomic what? Missing something here.
Line 381: “At that time…”. What time? There are four documents from 2004-2020.
Is Vietnam a “case for future research”?
It seems to be an ambition from the authors to convince the reader that the regulatory matters in Vietnam are of special interest. I am not convinced.
Some characteristics used in the text about the Vietnam system:
Line 390: “Generally, legal documents are scattered…”
Line 501: “It is not clear what management model in permitted areas.”
Line 502: “The promulgated documents have low legal value, are sporadic, and lack synchronization among existing legal documents.”
Line 513: “..competent authorities often avoid or are confused when dealing with violations of drone owners..”
Line 562: “…legislation is still unclear as to how it regulates smaller civilian unmanned aircraft, which raises the potential consequences regarding legal aspects, such as the inefficiency of compensation responsibility, violation of drone regulations, and information insecurity.
I first have to ask myself; is this a system in the front of development and which whom I want to learn from? Or if I accept the above statements, it looks more like a system not in the front of development. What, then, will research learn from Vietnam? How the development will be using a system with the above mentioned characteristics in a negative perspective? Or in a positive perspective, like “we may do very well without strict rules”? This is not clear to me.
Language and clarity.
The document would still need extensive English editing as multiple phrases and statements are not fully understandable with respect to what the authors intend to state.
Conclusion.
I still conclude that this paper does not fulfill crucial criteria for scientific articles, and I cannot see that it adds information of interest. I therefore cannot advise that this paper is accepted for publication.
Author Response
Dear reviewer,
The authors would like to thank the reviewer for the comments, which significantly helped to improve the manuscript. We have addressed all the issues raised in the review. For each comment, we provided a response and also described how the manuscript was revised. Those changes are highlighted within the manuscript. Please see attached file for our point-by-point response to the reviewer’s comments.

Reviewer 4 Report
After improving the paper it can be publish,
but in my opinion it's still "review" type of manuscript.
Author Response
Dear reviewer,
We would like to thank you again for the reviewer's efforts kindly.
Some essential aspects of this paper were improved, and the overall comprehensibility was enhanced.
Reviewer 5 Report
no more comments
Author Response

(The authors gave the same response as above.)
